# McArdle Disease: New Insights into Its Underlying Molecular Mechanisms

**DOI:** 10.3390/ijms20235919

**Published:** 2019-11-25

**Authors:** Francisco Llavero, Alazne Arrazola Sastre, Miriam Luque Montoro, Patricia Gálvez, Hadriano M Lacerda, Luis A. Parada, José Luis Zugaza

**Affiliations:** 1Achucarro Basque Center for Neuroscience, Science Park of the Universidad del País Vasco/Euskal Herriko Unibertsitatea (UPV/EHU), 48940 Leioa, Spain; alazne.arrazola@ehu.eus (A.A.S.); miriamluquem@gmail.com (M.L.M.); joseluis.zugaza@ehu.es (J.L.Z.); 2Faculty of Sports Science, European University of Madrid, 28670 Madrid, Spain; 3Department of Genetics, Physical Anthropology, and Animal Physiology, Faculty of Science and Technology, UPV/EHU, 48940 Leioa, Spain; 4Pharmascience Division, Technological Park of Health Sciences, Avda. de la Ciencia, s/n 18100 Armilla, Granada, Spain; p.galvez@bioiberica.com; 5Three R Labs, Science Park of the UPV/EHU, 48940 Leioa, Spain; hadriano101@icloud.com; 6Instituto de Patología Experimental, Universidad Nacional de Salta, 4400 Salta, Argentina; lparada@unsa.edu.ar; 7IKERBASQUE, Basque Foundation for Science, 48013 Bilbao, Spain

**Keywords:** McArdle disease, glycogen storage disease type V, glycogen phosphorylase, small GTPases, hexosamine biosynthetic pathway, *O*-glycosylation

## Abstract

McArdle disease, also known as glycogen storage disease type V (GSDV), is characterized by exercise intolerance, the second wind phenomenon, and high serum creatine kinase activity. Here, we recapitulate *PYGM* mutations in the population responsible for this disease. Traditionally, McArdle disease has been considered a metabolic myopathy caused by the lack of expression of the muscle isoform of the glycogen phosphorylase (PYGM). However, recent findings challenge this view, since it has been shown that PYGM is present in other tissues than the skeletal muscle. We review the latest studies about the molecular mechanism involved in glycogen phosphorylase activity regulation. Further, we summarize the expression and functional significance of PYGM in other tissues than skeletal muscle both in health and McArdle disease. Furthermore, we examine the different animal models that have served as the knowledge base for better understanding of McArdle disease. Finally, we give an overview of the latest state-of-the-art clinical trials currently being carried out and present an updated view of the current therapies.

## 1. General Characteristics of McArdle Disease

McArdle disease, also known as glycogen storage disease type V (GSDV; MIM#232600), is a severe form of glycogen storage disorder. It is an autosomal recessive disease caused by mutations in the gene encoding for the muscle isoform of glycogen phosphorylase (*PYGM*) (chromosome 11q13 gene) [1,2,3]. Glycogen phosphorylase (PYG) is the enzyme that catalyzes the first step of glycogenolysis to release glucose-1-phosphate (G1P) monomers from the intracellular glycogen stores [1].

This disease is included within the rare diseases category, and although the exact prevalence is not known, it has been estimated to be 1 in 100,000–140,000 patients [4,5]. The first symptoms occur during childhood and manifest as a syndrome of intolerance to exercise, cramps, fatigue, and muscle weakness. In principle, this pathology does not change the life expectancy of those affected [6]. In half of the patients, a massive increase in creatine kinase and rhabdomyolysis with myoglobinuria (dark urine) has been detected, which can lead to acute renal failure after exercising. Moreover, a “recovery” phenomenon known as the second-wind phenomenon is observed in many patients with relief of myalgia and fatigue after a few minutes of resting [7,8]. This clinical picture is usually standard, but some patients may manifest moderate or severe forms. For example, some cases have been described where the onset of the disease occurs at a very early age with hypotonia, generalized muscle weakness, and progressive respiratory failure [6].

The diagnosis of McArdle disease is based on biological findings that reveal a lack of elevation of blood lactate levels during the forearm ischemic exercise test, excess of glycogen, and deficit of myophosphorylase activity in the muscle biopsy. However, de Luna et al. have suggested that analysis of myophosphorylase expression in white blood cells might be a useful, less invasive, complementary test for diagnosing McArdle disease [9].

There is currently no specific treatment against McArdle disease, but approaches to it mostly involve treating the symptoms and avoiding the performance of intense physical exercise. Adjuvant therapy is based on performing a controlled physical exercise to develop mitochondrial oxidation capacities in muscles and glucose intake proportionally to the periods of exercise. On the other hand, ketogenic and protein-rich diets only had beneficial effects when the patients had already suffered an episode of rhabdomyolysis [5,10,11]. 

## 2. Glycogen Phosphorylase: Structure, Function, and Regulation

PYGM participates actively in the metabolism of carbohydrates by acting on intracellular glycogen stores; this is, therefore, a key enzyme to regulate glucose homeostasis and energy plasticity. Structurally, it is a homodimer protein of 97.4 KDa, and it is associated with the coenzyme pyridoxal phosphate (PLP or Vitamin B6) (Figure 1). Three isoforms constitute the family of PYG: the brain (PYGB), the liver (PYGL), and the muscle (PYGM) isoforms [12,13]. The brain isoform is predominant in the early stages of embryonic development and fetal tissues. From the postnatal stage, it is partially or completely replaced by the other two tissue-specific isoforms [14,15].

PYG activation can be regulated by two mechanisms: reversible phosphorylation and allosteric regulation [13]. PYGL can only be regulated by reversible phosphorylation at Serine 15 (S15), whereas PYGB and PYGM can be regulated by both serine phosphorylation and allosteric changes [16,17].

Reversible phosphorylation, taking place mainly in the liver, occurs in response to hormones such as glucagon, insulin, or adrenaline. The phosphorylase kinase (PK) is responsible for PYG phosphorylation, and the protein phosphatase 1 (PP1) is responsible for its dephosphorylation. Changes in the phosphorylation state induce conformational changes that rearrange the protein so that the catalytic region can bind to the substrate. When the PYG is phosphorylated on Serine 15, it changes to a conformational state known as R (relaxed with high affinity for the substrate), which is catalytically very active. By contrast, when the enzyme is dephosphorylated, it changes to a T state (tension state with low affinity for the substrate) that is inactive [16,17,18] (Figure 2).

In the 1950s, Sutherland described that both adrenaline and glucagon induced glycogenolysis in the liver. These hormones bind to G-protein coupled receptors (GPCRs), leading to the beta-gamma subunit (Gβ and Gγ) dissociation from the G-alpha (Gαs) subunit, which in turn activates the adenylyl cyclase (AC) [19]. The accumulation of intracellular cAMP causes the activation of protein kinase A (PKA), which causes the activation of PK [16,17,19,20,21,22,23]. In turn, PK activates PYG by phosphorylating its Serine 15. The signaling pathway constituted by GPCRs–AC–cAMP–PKA–PK leading to PYG activation is considered the classical or canonical pathway to glycogen phosphorylase activation.

Furthermore, AC activation can occur independently of GPCRs. Different isoforms of AC can be activated by both Ca^2+^ and direct phosphorylation due to the action of PKA and/or protein kinase C (PKC) [24], or Raf-1 [25,26]. In this fashion and in regard to the canonical pathway of glycogen phosphorylase activation, Llavero et al. have recently described the relationship between the epidermal growth factor receptor (EGFR) and glycogen phosphorylase activation. Specifically, this signaling pathway outlines a novel regulation of the canonical pathway adenylate cyclase 6 (ADCY6)–PKA–PK–PYG in which a receptor with intrinsic tyrosine kinase activity connects to the canonical pathway through the RAS–EPAC2–RAP1–RAF1 axis in a GPCR-independent manner. [18,27].

On the other hand, the allosteric regulation is produced by the binding of regulatory molecules to PYG, and these interactions produce the conformational changes of PYG that lead to the reversible activation or inactivation of glycogen phosphorylase. The allosteric regulation can be understood in terms of an equilibrium between the T and the R states mentioned above [12,16,28,29,30].

The allosteric regulators described by Fletterick et al. include the enzyme activators AMP, Pi, G1P, and glycogen, and the enzyme inactivators adenosine triphosphate (ATP), glucose-6-phosphate (G6P), glucose, and purine. The positive regulators bind to PYG, and their effects on its structure make the enzyme switch to the T or R state. Specifically, Pi and G1P bind to the active site of PYG but only activate it weakly, whereas the other activator, glycogen, opens the active site of PYG by binding on its N-terminal domain [31]. 

In the case of ATP and G6P (negative regulators), they bind to the PYG–AMP binding site, blocking the activation of PYG by AMP. Glucose binds to the active site disordering the N-terminal domain, and purine has the same effect as glucose, but it binds to the inhibitor site located near the active site [31,32].

Arrizabalaga et al. have recently reported that in addition to AMP, the small GTPase Rac1 is another novel allosteric activator of PYGM. The Rac1 active form (Rac1-GTP) binds to PYGM through the amino acids 181–317, and PYGM becomes active in human T cells (Figure 1). The integrity of this region of PYGM is therefore critical for its activation and IL-2-dependent lymphocyte proliferation [33]. Successive studies described that the main biological role of the Rac1-GTP/PYGM interaction is to control T-cell migration and proliferation [34,35].

Further, Llavero et al. recently linked the small GTPases of the Ras superfamily with PYG activation in an AC-dependent and -independent manner [18]. We postulate that the activation of one mechanism over the other is determined by different stimuli. When IL-2 is the stimulus, Rac1 is activated to modulate glycogen phosphorylase activation via allosteric changes. On the other hand, when the ligand epidermal growth factor (EGF) activates its specific receptor EGFR, Ras and Rap are activated, and then they phosphorylate Raf1 to stimulate glycogen phosphorylase activity [27,34,35]. The different signal transduction pathways activating PYG may be involved in different cellular responses. For example, whereas IL-2R controls T lymphocyte migration and proliferation through the Rac1/PYG pathway, EGFR signaling through the ADCY6/PYG pathway could be a mechanism to balance IL-2R activity and return T lymphocytes to their inactive state [18,27].

From a functional point of view, there are substantial differences amongst the three PYG isozymes. The glycogen phosphorylase muscle and brain isoforms act on the glycogen deposits to generate G1P. G1P is the substrate of the phosphoglucomutase, which transform it into glucose-6-P (G6P) and hence direct the metabolic machinery towards the production of ATP to regulate cellular functions such as muscle contraction [36]. However, the liver isoform hydrolyzes glycogen from internal liver stores to release G1P to maintain the physiological glucose levels in the bloodstream [37]. Further, the activation mechanisms are different; the liver isozyme is only activated by reversible phosphorylation of S15, while the other two isoforms can also be regulated by allosteric mechanisms (Figure 3).

On the other hand, when a cellular system is activated, such as T cells, the concomitant activation of carbohydrate metabolism will respond to both the demand for energy and the synthesis of macromolecules, which will be used in the cell division process [33].

## 3. Genetics of McArdle Disease: Mutations and Their Implications

In McArdle disease patients, the *PYGM* gene (11q13) mutations inactivate the enzyme. The mutation hotspots are presented in the *PYGM* gene exons 1 and 17, but 50% of the cases described are nonsense mutations [38,39]. Even though many mutations have been described, no correlation has been found yet between any mutation in each genotype and a specific phenotype [5]. Different mutations appear to produce similar symptoms. A total of 147 pathogenic mutations and 39 polymorphisms have been reported, with the arginine 50 to STOP (*p.R50X* or *R50X*) mutation being the most common [39]. This mutation represents about 40% to 50% of the alleles in McArdle disease patients in the Caucasian population, although in Asian populations, the *p.R50X* mutation has not been reported yet [39].

All these known mutations and polymorphisms have been identified by different studies. In one of them, three-point mutations were identified in the *PYGM* gene among 40 patients with McArdle disease [40]. Thirty-three patients were adults with the characteristic symptoms of the disease and six were children, including three siblings, and one infant [39]. Eighteen patients of the thirty-three analyzed, including the infant, were homozygous for the same nonsense mutation, *p.R50X*, originally reported as *R49X*. Twelve patients had a heterozygous *R50X* allele paired with another mutation in the *PYGM* gene. Hence, the *R50X* mutation was present in 75% of the patients. The last two patients analyzed were a family with apparent autosomal dominant inheritance: The mother was a compound heterozygote and the asymptomatic father carried another different mutation [41].

A DNA mutation analysis by restriction fragment length polymorphism (RFLP) of 54 Spanish (40 families) GSDV patients has shown that 78% of the mutant alleles were *R50X* and glycine 205 to serine *G205S*), originally reported as glycine 204 to serine *G204S*) and tryptophan 797 to arginine (*W797R*). It also identified six novel mutations in the *PYGM* gene but could not make any clear genotype-phenotype correlations [42]. Another study performed by Wu et al. identified other pathogenic mutations studying five unrelated McArdle patients. They identified an integrated heterozygosity consisting of the common R50X mutation and another pathogenic mutation in the *PYGM* gene (aspartic acid to glycine (*D51G*)). A sixth patient was homozygous for a small deletion [43].

In a study where ninety-four patients were analyzed (all Caucasians), around 55% of the mutated alleles had the most common *PYGM* pathogenic mutation *p.R50X*, whereas *p.W798R* and *p.G205S* accounted for 10% and 9% of the allelic variants, respectively. Seven new mutations were identified: *p.H35R*, *p.R70C*, *p.R94Q*, *p.L132WfsX163*, *p.Q176P*, *p.R576Q*, and *c.244-3_244-2CA*. Almost all patients showed exercise intolerance, the second wind phenomenon, and high serum creatine kinase activity. Furthermore, all the mutation analyses suggested there are no associations between *PYGM* genotype and the phenotypic manifestation of the disease [5].

## 4. PYGM Expression in Other Tissues

As was mentioned above, there are three glycogen phosphorylase isoforms expressed in humans: brain (PYGB), liver (PYGL), and muscle (PYGM). However, the predominant expression of an isoform in a specific tissue does not mean that this isoform is not present in other tissues [33,44,45,46,47,48]. The presence of one or more isoforms of glycogen phosphorylase in a tissue prompts the question about their specific roles in cell physiology.

Myophosphorylase or PYGM is mainly expressed in muscle; however, PYGM expression has also been detected in rat astrocytes, together with PYGB. Both isoforms co-localize perfectly in astrocytes both in the brain and spinal cord [46,47,49,50]. Moreover, presence of the PYGL isoform mRNA in cultured astrocytes suggests that this glial lineage is expressed in two or even all three isozymes at the same time [46,49,50]. All these findings suggest that each isoform will respond to different needs in astrocyte biology. For example, PYGM has been described to have a glycolytic supercompensation and glycogen shunt activity [46,49,50]. Further, Pinacho et al. postulated that the downregulation of Rac1 and PYGM could diminish the transfer of energy from astrocytes to neurons [47]. Schmid et al. also confirmed the expression of myophosphorylase in the kidney. The renal expression of PYGM was exclusively localized in interstitial cells of the kidney cortex and outer medulla, identified as fibroblasts [48]. Additionally, Arrizabalaga et al. demonstrated that Kit225 T cells express PYGM in addition to PYGL, with substantially higher expression of the former [27,33,34,35].

Furthermore, the retinal pigment epithelium (RPE) is another tissue reported to be affected in McArdle disease patients. Although PYGM expression still needs to be measured in these cells, four McArdle disease case reports with RPE dystrophy may indicate that this dystrophy can be related to PYGM mutations. Further, genetic screenings have demonstrated that these patients present mutations in the *PYGM* gene and not in the known dystrophy-causing genes, thus showing a possible relationship between retinopathy and McArdle disease [51,52].

Additionally, the results reported by Rodríguez-Gómez et al. suggest possible comorbidities with McArdle disease, as they show an undescribed condition in McArdle patients, who presented lower lean mass (LM) values in whole-body and regional sites, bone mineral content (BMC), and density (BMD) [53]. Further research needs to be done to understand the role of PYGM in this tissue. 

All these observations suggest that rather than affecting only the muscle, McArdle disease should be considered as a disorder that could affect many tissues such as the central nervous system (CNS), the immune, renal or bone systems. These findings may indicate that what are currently considered comorbidities may be tissues affected by the lack of myophosphorylase functional activity and suggest that McArdle disease is not just a neuromuscular disease (Figure 4). 

## 5. Glycogen Degradation: A Source for *O*-GlcNAcylation

In simple or complex multicellular organisms including humans, energy homeostasis is maintained through coordinated mechanisms amongst the various tissues and organs. Glycogen is the main source of energy in the muscle and liver [54,55]. In muscle, glycogen is used mainly as a fuel for ATP generation to supply energy for muscle contraction. By contrast, the role of glycogen in the liver is to maintain systemic glycemic homeostasis. On the other hand, glycogen function is not yet known in the rest of the tissues and organs [56]. For example, the role of glycogen in the CNS is a work in progress. It has been found that glycogen reservoirs have a high turnover rate that correlates with an increase in neuronal activity. Some functions on the role of glycogen in the CNS are related to memory formation [57] and to maintaining brain function during periods of low energy input [58].

Astrocytes have the highest glycogen concentration in the CNS. The glycogen of these cells can be mobilized by several molecules such as norepinephrine, intestinal vasoactive peptide, adenosine or high levels of potassium [59,60,61]. These molecules mediate the lactate generation from glycogen stores. Lactate has been extensively studied as a metabolic substrate, demonstrating that it is mobilized from astrocytes to neurons through monocarboxylate transporters (MCT) [62]. Lactate as a product of glycogenolysis has also been studied widely. For example, Perez-Escuredo et al. have shown that lactate is transferred through monocarboxylic transporters from stromal cells to cancer cells, as well as from cancerous cells with low energy demand (glycolytic cancer cells) to cells with high energy demand (oxidative cells) [63]. The functionality of lactate transfer has also been observed between the adipose tissue and the liver, playing a critical role in body weight [64]. 

Regarding glucose, this sugar can be released from intracellular glycogen stores, being both the fuel for cell energy metabolism and also the substrate for other pathways, such as the production of macromolecules (disaccharides, nucleotides or extracellular complex polysaccharides), or for the production of intermediary molecules of the hexosamine biosynthetic pathway (HBP).

It has been well established that the acute peak or an excess of nutrients in a short period leads to the activation of the HBP signaling cascade. The HBP transforms simple sugars such as glucose or fructose into the hexosamine uridine diphosphate N-acetylglucosamine (UDP–GlcNAc), which can be used for the *O*-GlcNAcylation of proteins [65]. Butkinaree et al. have shown that between 2% and 5% of intracellular glucose enters the HBP cascade. In this regard, Kang et al. observed that the exposure of cells to glucose deprivation increased the level of glycogen-dependent UDP-GlcNAc. This finding reveals that intracellular glycogen stores are required to increase the levels of *O*-GlcNAcylation under glucose deprivation and it suggests that cells release glucose from glycogen stores to maintain the intracellular ATP levels and also to increase the *O*-glycosylation levels of proteins [66].

Protein *O*-GlcNAcylation is a reversible post-translational modification (PTM) that occurs in serine and/or threonine residues of nuclear and cytoplasmic proteins [67]. It is a dynamic and ubiquitous process and it is responsible for the regulation of numerous biological processes. Recent glycoproteomic analyses taking into account that phosphorylation or *O*-GlcNAcylation PTMs occur in the same amino acidic residues (Ser, Thr) have shown that their levels are balanced in some proteins. This equilibrium serves as a nutrient sensor against different stresses to modulate the intracellular signaling cascades, protein transcription, and arrangement of cytoskeletal architecture [68]. There is increasing evidence showing that *O*-GlcNAcylation levels rise in response to stress and that this sharp increase is cytoprotective, at least in the short term [68,69,70]. By contrast, a reduction in *O*-GlcNAcylation levels seems to be associated with a decrease in cell survival in response to acute stress [71]. In fact, the *O*-glycosylation/phosphorylation balance controls important signaling pathways involved in cell physiology [72], such as the NF-κβ pathway [73,74,75], the tumor suppressor hypermethylated in cancer 1 (HIC1), and c-Myc [68]. Hsieh-Wilson’s team demonstrated the great relevance of this type of PTM in the regulation of metabolic cascades. Specifically, they observed that the inhibition of *O*-glycosylation in S529 of phosphofructokinase1 (PFK1), in addition to reducing cancer cell proliferation in vitro, also impairs tumor formation in vivo [76]. Regarding the implication of glycosylation in tumor processes, it is very well established that this PTM plays an essential role in breast, lung, and colon tumors [73,74,77,78]. Moreover, it has been reported that abnormal levels of glycosylated proteins underlie pathologies such as neurodegenerative diseases, type II diabetes, cancer, and, as recently described, AIDS [79].

The activity of the PYGM together with the G1P on glycogen breakdown perform an important role in both tissue functionality and cell biology. However, it could be postulated that in McArdle disease, the lack of PYGM activity could affect not only ATP generation, but also PTM processes, altering the *O*-GlcNAcylation of some proteins and affecting other tissues than skeletal muscle, such as the immune system and/or the brain (Figure 3). 

## 6. Animal Models for the Study of McArdle Disease

Different animal models were developed to understand what the genetic and molecular bases of McArdle disease are. The first animal model of McArdle disease was described by Angelos et al., who identified a natural and spontaneous myophosphorylase deficiency in a Charolais bovine race [80]. This study established that the bovine amino acid sequence and nucleotide sequence homology to the human PYGM is 95.8% and 92.0%, respectively. Moreover, the substitution of C to T at codon 489, changing the coded arginine (CGG) to tryptophan (TGG) resulting in an inactive myophosphorylase, was identified by quantitative PCR. This mutation is adjacent to pyridoxal phosphate binding sites, a key region for the myophosphorylase catalytic activity. Furthermore, this mutation is highly conserved in different species [40,80,81].

In 1997, a unique herd of Merino sheep at the farm of the Veterinary School of the University of Murdoch in Western Australia was described as PYGM-deficient [82]. The mutation occurs in the 3′ acceptor splice-site of intron 19 of the ovine PYGM gene, resulting in the activation of a cryptic splice-site in exon 20 and the premature termination of the transcript [83]. These sheep lack glycogen phosphorylase activity in the skeletal muscle and show the same clinical effects and morphological changes to those described in human patients [44,84]. The PYGM-deficient sheep model is the first animal model in a species with a lifelong body mass similar to that of humans. This is an unique animal model with huge value for performing preclinical tests. In addition, it is also relevant for the evaluation of both benefits and risks of the therapies used. Moreover, the experimental research developed in these sheep has led to human clinical trials [44,84]. These studies are based on intramuscular injections on the surface of the sheep’s muscle, giving rise to an increase in the strength and the regeneration of muscle fibers [44,84]. In addition to the effect of sodium valproate on skeletal muscle, PYGM expression in the McArdle disease sheep has also been examined, and the findings suggest that this compound could be a potential candidate for McArdle disease therapy [84]. Although two spontaneous animal models have been identified for McArdle disease (Charolais cattle and sheep), they have provided limited information over the pathophysiology of this disease, since they are not good experimental animals due to their long reproductive cycle, which complicates the obtaining of developed animals for replicating the experiments and reproducing the results.

In this way, a ‘knock-in’ mouse was generated by replacing the wild-type PYGM allele with a modified one carrying the most frequent mutation in McArdle patients (p.R50X) [85]. These homozygous and wild-type mice were subjected to a wide variety of phenotypic studies, including immunohistochemical and biochemical, as well as exercise tests. This experimental model represents a tool for in-depth studies of the pathophysiology of McArdle disease, as well as to explore new therapeutic approaches [85]. 

In addition to examining the effects of physical exercise on the knock-in *p.R50X* mice [86], investigation of some of the signaling pathways in this animal model cells demonstrated that the lack of PYGM expression causes alterations in sensory energetic cascades together with some evidence of oxidative damage and, at the same time, alterations in Ca^2+^ flux but without major alterations in the oxidative phosphorylation capacity or the autophagy/ubiquitination pathways. These results suggest that the muscle tissue of McArdle disease patients is probably suitable to moderate sessions of physical exercise [87]. Subsequently, it was observed that the p.R50X mice’s muscles seem to adapt to the energy deficiency. In p.R50X mice, there is an increase in the expression and activation of proteins like insulin receptor, 5’ adenosine monophosphate-activated protein kinase, Akt, and hexokinase II. All these proteins are involved in the metabolism of blood-glucose uptake in response to exercise. There have also been unsuccessful attempts to obtain cell lines from McArdle patients. To overcome this obstacle, Birch et al. generated cellular models expressing the wild-type or mutant PYGM (R50X or G205S) by integrating cDNA into the genome of Chinese hamster ovary cells (CHO cells) [88].

## 7. Clinical Trials in McArdle Disease

Regarding the clinical trials carried out to completely or partially alleviate the symptoms of McArdle disease, different food supplements have been tested, such as ribose together with physical exercise as coadjutant therapy [89], or the dietary oil Triheptanoin, (ClinicalTrials.gov Identifier: NCT02432768). Similarly, sodium valproate has also been tested as a treatment for McArdle disease. Sodium valproate is an anticonvulsant medication belonging to a group of drugs known as histone deacetylase inhibitors (HDACs), which regulate gene expression. Sodium valproate treatment in an ovine model of McArdle disease showed an increase in the expression of a PYGB [84], and primary cultures of myocytes isolated from McArdle knock-in mice showed that sodium valproate increased PYGB expression and reduced the accumulation of intracellular glycogen [90]. However, the administration of sodium valproate was of no benefit to McArdle disease patients [84] (ClinicalTrials.gov Identifier: NCT03112889). 

Other clinical trials were conducted with a potential diet therapy of modified ketone bodies. Ketone bodies or ketones are fat waste products. They occur when the body uses fats instead of sugars to generate ATP. Ketone bodies are formed in situations where glucose metabolism is compromised. Ketosis can be reached by fasting or it can be induced by adhering to a modified ketogenic diet, which entails a high-fat, low-carbohydrate diet, which simulates the metabolic effects of fasting. In these trials (NCT03843606 and NCT04044508), it was postulated that a modified ketogenic diet could be a potential treatment option, by providing ketones as alternative fuel substrates for the working muscle, since McArdle patients are unable to utilize sugar stored as glycogen in muscles. Ketosis can potentially provide alternative fuel substrates by providing endogenous ketone bodies (KBs), which are acceptable fuels for skeletal muscle and brain. The chemical compounds are acetoacetic acid (acetoacetate) and beta-hydroxybutyric acid (β-hydroxybutyrate); one part of the acetoacetate undergoes non-enzymatic decarboxylation giving acetone (an insignificant amount under normal conditions); the first two are acidic and the third, a ketone. A part of acetoacetate is reduced to β-hydroxybutyrate in the mitochondria itself, which consumes an equivalent of ATP (a molecule of NADH). 

An additional trial will investigate the immediate effects of the oral supplementation of exogenous ketone bodies (food supplement containing β-hydroxybutyrate esters) on the exercise capacity in patients with metabolic myopathies, compared with a placebo drink (NCT03945370). These studies are ongoing.

In 2002, a study showed that the administration of creatine supplementation improved work capacity in patients with McArdle disease. They also assessed the efficacy of creatine therapy in McArdle disease [91,92]. Subsequent studies reported that the effects of creatine on the muscles may be independent of their energy metabolism. In one case of ketogenic diet, there was improvement in muscle symptoms and performance. However, again, these effects did not result in visible changes in muscle energy metabolism [93]. 

Lastly, in a few published randomized controlled trials in McArdle disease, oral administration of a low dose of creatine affords a modest benefit in ischemic exercise and in a small number of patients [91,92,93], while oral administration of sucrose before planned exercise improved patient performance [94,95,96]. Furthermore, there was low-quality evidence of improvement in some of the parameters with creatine, oral sucrose, Ramipril, and carbohydrate treatment. However, these treatments are not a solution for McArdle patients’ everyday living. Other than aerobic exercise, it is still not possible to recommend any specific treatment for McArdle patients [94].

## 8. Future Perspectives

Regarding the diagnostic detection of McArdle disease, de Luna et al. recently made a significant advance when they proposed that the expression and genetics analysis of *PYGM* could be made in white blood cells instead of the classical muscle biopsy, being infinitely less invasive for patients and more economic for national health systems [9].

This observation could point out a new paradigm regarding McArdle disease, to move from a disease affecting only the skeletal muscle to a syndrome affecting many tissues. In this way and taking into account that glycogen serves as a source for UDP–GlcNAc, which is used as a substrate for *O*-glycosylating proteins, the effect of these PTMs should be investigated to understand how their function is altered in McArdle disease patients. Thus, the characterization of glycosylated proteins would open a new venue and expectations in McArdle disease therapy.

In the future, a potential therapeutic solution could be focused on gene therapy. This approach could allow health professionals to treat the disorder by inserting the wild-type *PYGM* gene into the patient’s cells. Although gene therapy is a promising treatment option for several diseases (including inherited disorders, some types of cancer, and certain viral infections), the technique remains risky and is still under basic investigations to make sure that it is safe and effective.

Finally, a major problem of therapeutic studies for McArdle disease is the low prevalence of the disease. A possible improvement would be designing clinical trials with a large number of subjects, taking consortiums, like a registry of patients with McArdle disease (EUROMAC), as a reference [97]. Thus, creating a big cooperative multicenter and multinational studies would be a good option. Further, there is a need to develop a generic database with outcome data, including baseline parameters in a large cohort of subjects, as previously proposed. This would generate a valuable knowledge-database on which to rely for the holistic interpretation of results, in addition to being a great platform to design future assays. This would help to obtain results that not only provide a mechanistic interpretation but also serve to improve the quality of life of McArdle patients.

## Figures and Tables

**Figure 1 ijms-20-05919-f001:**
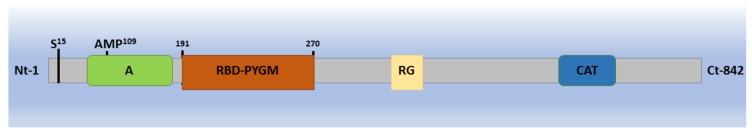
The muscle phosphorylase isoform (PYGM) sequence presents the following information: Serine 15; A, allosteric region; AMP, adenosine monophosphate (AMP) interaction site; RBD-PYGM, Region binding domain of PYGM to Rac1-GTP that exhibits significant homology with the Rac binding domain of PAK1 (serine/threonine p21-activated kinase); RG, glycogen reserve region; CAT, catalytic region; Nt, amino terminal; Ct, carboxyl terminal.

**Figure 2 ijms-20-05919-f002:**
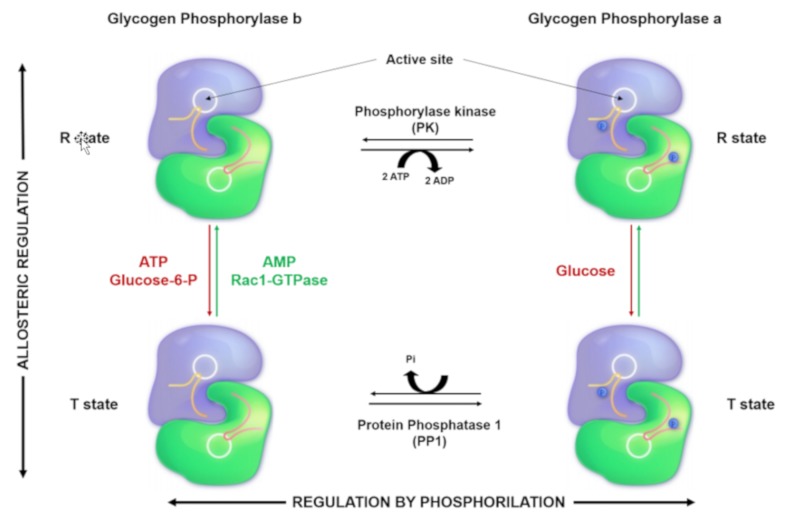
PYGM regulation: Glycogen phosphorylase (PYGM) is regulated both by phosphorylation (X axis) and by allosteric mechanisms (Y axis). The phosphorylation of Serine 15 (S15) of the amino terminal induces the change in the PYG homodimer in its inactive form (b) to its active form (a). Both forms are in a balance between the tense state (T) and the relaxed state (R). Although the equilibrium favors the T state of the b form and the R state of a form, the molecules responsible for allosteric activation of PYG b are adenosine monophosphate (AMP) and GTPase Rac1 (Rac1GTPase), which bind to their respective allosteric regions. The union of AMP or the phosphorylation of the S15 induces a conformational change, freeing the entrance to the catalytic region, which is hindered in the state tense. The enzyme responsible for phosphorylating S15 of PYG is phosphorylase kinase (PK), and protein phosphatase 1 (PP1) is responsible for dephosphorylating it. (Adapted from Bardford and Johnson 1989, Nature; Johnson 1992, Protein Science).

**Figure 3 ijms-20-05919-f003:**
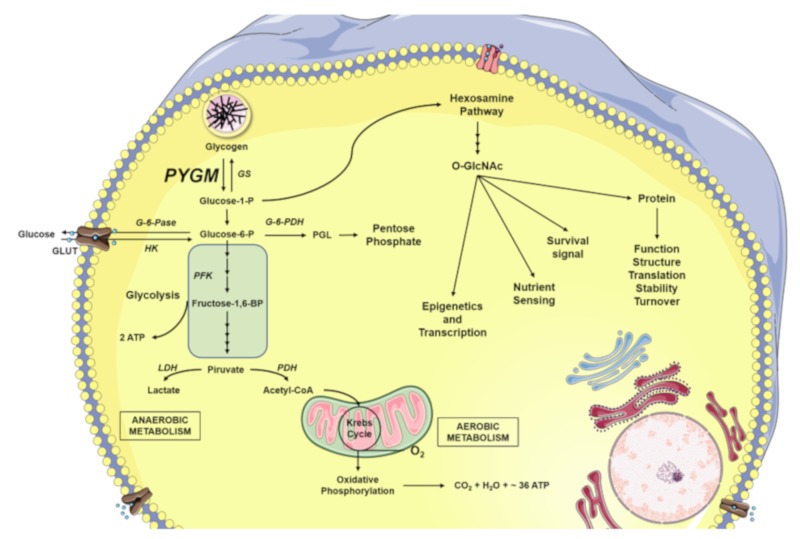
Glucose metabolism. Both glucose-1-phosphate (G1P) released from the intracellular glycogen stores by glycogen phosphorylase (GP), as well as the glucose introduced into the cell are converted to glucose-6-phosphate (G6P) by phosphoglucomutase (PGM) and hexokinase (HK), respectively. The G6P can be directed to different destinations. One of them is the pentose phosphate pathway for the formation of nucleic acids, which is directed by the G-6-P dehydrogenase (G-6-PDH), forming the first product of 6-phosphoglucose δ-lactone (PGL). Another destination can be glycolysis, in which the phosphofructokinase (PFK) plays a primordial role, producing fructose 1,6-bisphosphate (F-1,6-BP); the glycolytic reactions will culminate in the production of pyruvate and adenosine triphosphate (ATP). Pyruvate can be fermented in lactate by the catalysis of the lactate dehydrogenase (LDH), as happens during anaerobic muscle exercise. On the other hand, pyruvate can be used to obtain ATP through oxidation. The first enzyme that participates in this chain of reactions is the pyruvate dehydrogenase (PDH), which produces Acetyl CoA, a substrate of the Krebs cycle, which together with oxidative phosphorylation produces molecules of ATP, carbon dioxide (CO_2_), and water (H_2_O). Glucose, moreover, in addition to being the fuel of the cell’s energy metabolism, is also used by the cellular machinery as a vitally important substrate for the production of key intermediaries of the hexosamine biosynthetic pathway (HBP). GS, glycogen synthase; *O*-GlcNAc, β-linked *N*-acetylglucosamine; GLUT, Glucose transporters.

**Figure 4 ijms-20-05919-f004:**
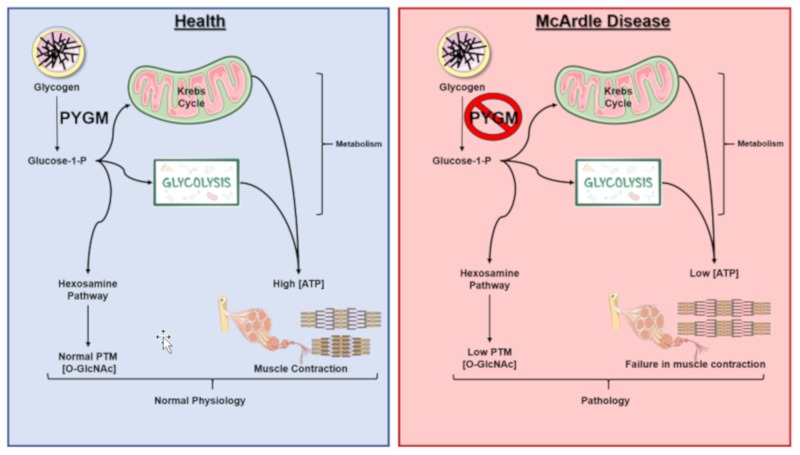
McArdle disease Scheme. McArdle disease, also referred to as myophosphorylase deficiency or type V glycogen storage disease, is a recessive inherited disorder characterized by an inability to metabolize glycogen due to the absence of a functional myophosphorylase (PYGM). Patients lack sufficient glucose-1-phosphate (G1P) monomers needed for glycolysis and the hexosamine biosynthetic pathway (HBP). This results in lower ATP and, consequently, lower muscle contraction, as well as in lower post-translational modifications by *O*-GlcNAcylation in comparison to normal conditions. PTM: post-translational modification; *O*-GlcNAc, β-linked *N*-acetylglucosamine.

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
