# Peer review of "McArdle Disease: New Insights into Its Underlying Molecular Mechanisms"

_ijms, 2019, doi:10.3390/ijms20235919_

Round 1

Reviewer 1 Report

The authors made significant changes.

Author Response

This reviewer indicates that:

Point 1

In general, make sure that where appropriate ‘energy’ is replaced by ‘ATP’ as one is not able to ‘generate energy’ as it is rather transforming energy from one form to another.

Response point 1: We agree, we have corrected it.

Point 2

What is meant by:

Line 493: The PYGM-deficient sheep 492 model presents a life-long body mass similar to that of humans.

Response point 2: We agree, we have replaced it for: “The PYGM-deficient sheep model is the first animal model in a species with lifelong body mass similar to that of humans.”

Point 3

Why are they:

Line 519: since they are not good experimental animals

As they even led to human clinical trials!

Response point 3: We agree, we have substituted it for: “they have provided limited information over the pathophysiology of this disease since they are not good experimental animals due to their long reproductive cycle, which complicates the obtaining of developed animals for replicating the experiments and reproducing the results.”

Point 4

Explain all abbreviations in figures in the legend.

Response point 4: We agree, we have incorporated it the following abbreviations:

In the Figure 1 (AMP, adenosine monophosphate (AMP) interaction site; RBD-PYGM, Region binding domain of PYGM to Rac1-GTP that exhibits significant homology with the Rac binding domain of PAK1 (serine/threonine p21-activated kinase).

In the Figure 3 (phosphoglucomutase (PGM); hexosamine biosynthetic pathway (HBP). GS, glycogen synthase; O-GlcNAc, β-linked N-acetylglucosamine; GLUT, Glucose transporters)

In the Figure 4 (myophosphorylase (PYGM). glucose-1-phosphate (G-1-P) hexosamine biosynthetic pathway (HBP); O-GlcNAc, β-linked N-acetylglucosamine)

Reviewer 2 Report

In general, make sure that where appropriate ‘energy’ is replaced by ‘ATP’ as one is not able to ‘generate energy’ as it is rather transforming energy from one form to another.

What is meant by:

Line 493: The PYGM-deficient sheep 492 model presents a life-long body mass similar to that of humans.

Why are they:

Line 519: since they are not good experimental animals

As they even led to human clinical trials!

Explain all abbreviations in figures in the legend.

Author Response

This reviewer indicates that:

Point 1

In general, make sure that where appropriate ‘energy’ is replaced by ‘ATP’ as one is not able to ‘generate energy’ as it is rather transforming energy from one form to another.

Response point 1: We agree, we have corrected it.

Point 2

What is meant by:

Line 493: The PYGM-deficient sheep 492 model presents a life-long body mass similar to that of humans.

Response point 2: We agree, we have replaced it for: “The PYGM-deficient sheep model is the first animal model in a species with lifelong body mass similar to that of humans.”

Point 3

Why are they:

Line 519: since they are not good experimental animals

As they even led to human clinical trials!

Response point 3: We agree, we have substituted it for: “they have provided limited information over the pathophysiology of this disease since they are not good experimental animals due to their long reproductive cycle, which complicates the obtaining of developed animals for replicating the experiments and reproducing the results.”

Point 4

Explain all abbreviations in figures in the legend.

Response point 4: We agree, we have incorporated it the following abbreviations:

In the Figure 1 (AMP, adenosine monophosphate (AMP) interaction site; RBD-PYGM, Region binding domain of PYGM to Rac1-GTP that exhibits significant homology with the Rac binding domain of PAK1 (serine/threonine p21-activated kinase).

In the Figure 3 (phosphoglucomutase (PGM); hexosamine biosynthetic pathway (HBP). GS, glycogen synthase; O-GlcNAc, β-linked N-acetylglucosamine; GLUT, Glucose transporters)

In the Figure 4 (myophosphorylase (PYGM). glucose-1-phosphate (G-1-P) hexosamine biosynthetic pathway (HBP); O-GlcNAc, β-linked N-acetylglucosamine)

This manuscript is a resubmission of an earlier submission. The following is a list of the peer review reports and author responses from that submission.

Round 1

Reviewer 1 Report

The authors give an overview of the literature related to McArdle’s disease.

The title suggests it will deal with the mechanism of McArdle disease. Indeed, the different mutations in the gene for myophosphorylase is the underlying cause, and many mutations are discussed, but I guess a more appropriate title would be something along the lines of ‘The molecular effects of McArdle’s disease’, as that seems to be the thrust of the review.

The review could do with a thorough check of the English, and a better construction, starting with what McArdle’s disease is, the structure and function of and regulation of the enzyme, then mutations followed by the potential effects of disturbed glycogen breakdown, the treatment options and limitations, ending with animal models and future directions. When you talk about trials you need also to explain, if that is known, what the rationale is for the trials. For instance, how would sodium valproate work in this disease.

Abstract:

The phrase ‘Molecular mechanism which participated on glycogen phosphorylase activation’ is rather vague. The sentence after this statement also does not tell me much, and the sentence after that is also rather vague. It all seem rather telegramme style sentences that don’t tell me anything about what happens during McArdle’s disease, or what the mechanism is. Please say what you mean.

Page 2 line 38: It does not imply it, but rather it is the absence of a functional myophosphorylase.

Line 41: not ‘=’ but rather ‘à’

Page 3 line 63: is an autosomal recessive ‘disease’.

Page 3 line 65: ‘Misunderstandings’?

Line 66-68: Strange sentence; what is it you want to tell the reader?

Line 72-73: Strange sentence.

Line 73-74: Sentence construction.

Line 83-84: Incomplete sentence.

Throughout the manuscript the writing needs considerable attention. You need to make clear what you want to tell the reader with each sentence, check the grammar, and check that the sentences make sense. Abbreviations need to be explained at first use.

Have the description of mutations after ‘structure, function and regulation’. That section reads quite well.

Figure 2: ‘Hexokinase’, rather than ‘hexoquinase’ Make sure the names of enzymes and substances in the figure are English.

The section ‘Glycogen physiology’ can be removed. Stay focussed on the subject of the review, which is McArdle’s disease.

The story of glycosylation is interesting, but only limited data are given, or studies discussed, on changes in glycosylation in McArdle’s disease.

Reviewer 2 Report

The manuscript entitled “Molecular Mechanism of McArdle Disease” summarized some progress in the studies associated with McArdle disease, which is a genetic glycogen storage disease due to the absence of glycogen phosphorylase. The manuscript provides some new insights, such as the functions of glycogen phosphorylase in other organs and tissues, for understanding the outcomes of McArdle disease. However, there are some important issues in this manuscript:

1. The content of the manuscript doesn’t address the title “molecular mechanism of McArdle disease” very well. For instance, there are two diagrams in the manuscript, but they are not providing clear information directly concerned with the pathways or signaling for the development of McArdle disease. Moreover, is it correct to use “no muscle contraction” in the right panel of figure 1? Furthermore, the legend of figure 1 doesn’t explain the figure very well.

2. Many references in the manuscript are published several decades ago, the authors should include more new information from current development of the research associated with McArdle diseases.

3. Since it is “Molecular Mechanism of McArdle Disease”, what could be the direct molecular mechanisms or pathways behind the development of the symptoms of McArdle disease? For example, the patients show less lean mass and lower bone mineral density, what could be the reasons? Why some mutations lead to the disease, but some do not have the same function? Can the animal models mimic the development of the disease in human? If not, why? The authors use several pages to explain glycogen physiology and glycogen degradation, but how these two parts are related with the symptoms or the development of McArdle disease?

4. what could be the rationale of previous or current clinical trials? This information is important for understanding the molecular mechanism of McArdle disease.

5. Currently, human skeletal muscle cells from healthy human subjects and patients are widely used in research, why it is not feasible to get muscle cells from McArdle patients?

6.More references should be provided in the part of “General characteristics of McArdle disease”.

Overall, the organization of the manuscript needs to be improved, more information directly related with the development or the symptoms of McArdle disease should be provided.